# Nanoelectromechanical Temperature Sensor Based on Piezoresistive Properties of Suspended Graphene Film

**DOI:** 10.3390/nano13061103

**Published:** 2023-03-19

**Authors:** Shuqi Han, Siyuan Zhou, Linyu Mei, Miaoli Guo, Huiyi Zhang, Qiannan Li, Shuai Zhang, Yaokai Niu, Yan Zhuang, Wenping Geng, Kaixi Bi, Xiujian Chou

**Affiliations:** 1School of Instrument Science and Technology, North University of China, Taiyuan 030051, China; 2Key Laboratory of National Defense Science and Technology on Electronic Measurement, School of Instrument and Electronics, North University of China, Taiyuan 030051, China; 3School of mechanical engineering, North University of China, Taiyuan 030051, China; 4School of Semiconductors and Physics, North University of China, Taiyuan 030051, China

**Keywords:** nano electromechanical, piezoresistive, graphene, temperature sensors

## Abstract

The substrate impurities scattering will lead to unstable temperature-sensitive behavior and poor linearity in graphene temperature sensors. And this can be weakened by suspending the graphene structure. Herein, we report a graphene temperature sensing structure, with suspended graphene membranes fabricated on the cavity and non-cavity SiO_2_/Si substrate, using monolayer, few-layer, and multilayer graphene. The results show that the sensor provides direct electrical readout from temperature to resistance transduction by the nano piezoresistive effect in graphene. And the cavity structure can weaken the substrate impurity scattering and thermal resistance effect, which results in better sensitivity and wide-range temperature sensing. In addition, monolayer graphene is almost no temperature sensitivity. And the few-layer graphene temperature sensitivity, lower than that of the multilayer graphene cavity structure (3.50%/°C), is 1.07%/°C. This work demonstrates that piezoresistive in suspended graphene membranes can effectively enhance the sensitivity and widen the temperature sensor range in NEMS temperature sensors.

## 1. Introduction

Graphene has attracted tremendous attention [1,2,3,4] due to its excellent mechanical strength [5,6,7,8], outstanding electrical properties [9,10,11], remarkable electrical conductivity [12,13], and significant temperature-sensitive properties [14,15]. Graphene temperature sensors have been extensively explored based on various principles, namely fiberoptic [16,17,18], strain [19,20,21,22], and thermal resistance [23,24,25]. The temperature sensor with fiberoptic is one of the most imperative graphene temperature sensors with high sensitivity, precision, and repeatability. For instance, Jun Zhang et al. [17] demonstrated an all-fiber-optic temperature sensor. The sensor’s sensitivity is 0.134%/°C, and the temperature range is −7.8–77 °C. Such sensors have some limitations about unsuitable in high-temperature and complex fabrication. Strain graphene temperature sensors have a mature manufacturing technology compared to fiber-optic sensors. M Saafi et al. [19] presented a high-precision temperature sensor with a high sensitivity of 1.98%/°C. But the sensor’s resistance change typically occurs in the narrow range of −10–35 °C, limiting their applications in wide-range temperature sensing. Besides, the thermal resistance graphene temperature sensor is the most widespread one. Its maximum detectable temperature is up to 2100 °C [26]. The corresponding sensitivity is higher than fiber-optic and strain graphene temperature sensors. Pengzhan Sun and co-workers [24] have exhibited a graphene temperature sensor with negative temperature coefficient resistance values of −1.41%/°C and −0.53%/°C at 28–45 °C and 50–160 °C. However, the substrate impurities scattering of graphene will lead to unstable temperature-sensitive behavior and poor linearity in graphene temperature sensors. Sometimes a change from a negative to a positive temperature coefficient of resistance is observed for graphene on SiO_2_ [27]. 

Some innovative works are carried out to solve the above challenges. For example, the preparation of suspended graphene on Si/SiO_2_ substrates was adopted to improve sensitivity [28]. Bolotin and co-worker’s work demonstrated that improving or eliminating the substrate by suspended graphene over a trench seems a promising strategy for higher sensitivity [28]. Jian-Hao Chen and co-worker’s experimental data also [29] revealed that the preparation of suspended graphene could effectively reduce the effects of substrate impurity scattering and electron-phonon scattering on the electrical parameters of graphene. The suspending graphene can effectively reduce the substrate impurity scattering on the electrical parameters of graphene [29]. However, thermal resistance is also weak (or even non-existent) in suspended graphene [30]. That is why sensors based on the thermal resistance of graphene do not use suspended graphene structures. In summary, the thermal resistance graphene temperature sensor suffers from the drawbacks of low sensitivity and the narrow temperature sensor range.

In this work, a cavity graphene structure (Figure 1a) on a silicon substrate will enhance the sensitivity and widen the temperature sensor range. A pressure difference is present between the inside and the outside of the cavity. And this is caused by a change in temperature (Figure 1c) [31]. As a result, the graphene membranes sealing in the caves are deflected and thus strained. Then the mechanical strain in the graphene induced a piezoresistive effect, which changes the electronic band structure [32]. This further leads to a change in device resistivity. The output resistance of the sensor in this work will increase with the temperature. Due to the piezoresistive effect, the substrate scattering effect and the thermal resistance effect all contribute to the output resistance of the sensor. In order to let the piezoresistive effect play a dominant role in the output resistance variation of the device. The suspended graphene (Cavity structure) and increasing the number of graphene layers to attenuate the effects of these two effects on the device output resistance. In a word, the cave structure uses a piezoresistive to complete temperature sensing when the cave structure makes graphene suspended (Figure 1b). Making graphene suspended not only can reduce the substrate impurity scattering on the electrical parameters of the graphene sensor but also avoid the influence of unstable thermal resistance on the electrical parameters of graphene. And this work also provides systematic research on temperature sensing including monolayer, few-layer, and multilayer graphene materials. This work can provide theoretical and technical support for NEMS temperature sensors.

## 2. Materials and Methods

### 2.1. Simulation

As shown in Figure 2a, the graphene directly above the cavity is in suspension, and the rest tightly adheres to the SiO_2_ surface. And the graphene will seal the gas in the cavity, which is led by excellent air tightness and van der Waals forces between graphene and SiO_2_. The gas pressure in the cavity increases as the temperature rises, leading to the gas pressure difference as shown in Figure 2b,c. This gas pressure difference causes the deformation of the suspended graphene, causing the piezoresistive effect. The piezoresistive of graphene will change the resistance of the suspended graphene film. It is predicted from the analysis in the introduction that the output resistance of the device will be changing with temperature. In this work, we simulate the range of air pressure in the cavity at different temperatures, the strain of graphene at different pressures and the piezoresistive effect of graphene.

The pressure strain of multilayer graphene with PMMA was simulated using COMSOL Multiphysics 5.5. In the simulation of air pressure in the cavity, we assume that gases are in standard atmospheric pressure. And the $p_{1}$ = 1.013 × 10^5^ Pa is the initial air pressure at the two sides of the cavity. The gas pressure formula is given as follows [33]:(1)$$\frac{T_{1}}{p_{1}}=\frac{T_{2}}{p_{2}}$$

Under the temperature sensing range of 25 °C–120 °C, the pressure range of suspended graphene is 0–3.84 × 10^5^ Pa. The size of the cavity is 1.5 mm × 1.5 mm × 50 μm, and the volume of the gas in the cavity is 1.12 × 10^−12^ m^3^. The range of gas pressure values in the cavity is converted to Newtonian mechanical units of 0.2 N–9.6 N. The atmospheric pressure on the suspended graphene is 0.2 N, and the pressure difference on the suspended graphene from the previous analysis is 0 N−9.4 N. The experimental conditions were chosen to remain within the expected tearing limits of the graphene membrane. It can be seen from SEM images that the maximum stretch of the graphene film reaches about 20% [31,34,35].

The PMMA layer in the device designed in this work protects the graphene. PMMA covering the graphene should be unavoidable, which is contrary to the argument. Therefore, the PMMA layer was considered in the simulation. The detailed parameter settings in the simulation are as follows. The thickness of suspended graphene film t = 10 nm, Young’s modulus E = 1 TPa, Poisson’s ratio γ = 0.16, The thickness of PMMA t_2_ = 500 nm, Young’s modulus E = 1.6 GPa, Poisson’s ratio γ = 0.22. The simulation results are shown in Figure 2b,c. The simulation results show that the maximum displacement of the suspended graphene is 8 nm.

If a pressure difference is present between the inside and the outside of the cavity (compare Figure 1c), the graphene membrane that is sealing the cavity is deflected and thus strained. This leads in a change of device resistivity due to the piezoresistive effect in the graphene [31]. The simulation of the piezoresistive effect of graphene will show the relationship between resistance and temperature. Nevertheless, further studies regarding stability are required in the experiment. And the electromechanical piezoresistive of graphene was stimulated in MATLAB. The piezoresistive effect of graphene is expressed by Equation (2) [23,36].
(2)$$R=\rho \frac{L}{W_{t}}=\frac{1}{2qN_{e}\mu_{e}}\cdot \frac{\left(1+\varepsilon_{xx}\right)}{(1+\varepsilon_{yy)}}\cdot \frac{L}{W_{t}}$$
where the ρ is the resistivity of the membrane region, $N_{e}$ is the electron density, and $\mu_{e}$ is the electron mobility. $\varepsilon_{xx}$ and $\varepsilon_{yy}$ are strain components in the x and y directions of the membrane region. Furthermore, q is the electronic charge, and L and $W_{t}$ are the change parallel and perpendicular to the stress of the graphene film. The simulation results show in Figure 2d that the output resistance of the temperature sensor will increase with the temperature rising.

Ignoring the substrate impurity scattering and the thermal resistance effect of graphene. The piezoresistive simulation shows that the device resistance will increase with rising temperature. Piezoresistive effects, substrate scattering effects, and thermal resistance effects all contribute to the output resistance of the sensor. In this work, piezoelectric effect simulations are performed to theoretically verify the role of strain and resistance of graphene. Therefore, the effect of other factors on the resistance is not considered in the simulation. Moreover, the thermal resistance effect of graphene is not the core of the sensor in this article. On the contrary, the thermal resistance effect of graphene is an effect that the paper hopes to eliminate. The substrate scattering effect of silicon substrates on graphene has a significant impact on the performance of the monolayer sensors in this work. However, there is no complete theoretical system for the research results on the scattering effect of silicon substrates on graphene [29]. Therefore, an effective simulation of the base-scattering effect is not available. As a matter of course, the effect of the substrate scattering effect on the sensor resistance is analyzed in the results section in conjunction with test data and existing theory. Nevertheless, the test data need to be analyzed in a combination of three effects.

### 2.2. Experiments

#### Device Fabrication

The CVD graphene on copper foil used in the device preparation process is transferred to the target substrate by wetting transfer technology. The PMMA is a protective layer for graphene during the wet transfer process. In addition, PMMA, a sacrificial layer in the wet transfer process, is not removed and continues to protect graphene during the temperature sensing process. The detailed: (a) Removal of residual organic matter from the back side of graphene copper foil using a glue remover. (b) The PMMA solution was spin-coated on the front side of the copper-based graphene and cured on a hot plate at less than 60 °C. (c) The cured copper-based graphene is put into the configured copper sulfate solution to etch the copper base. (d) After all the copper corroded, the graphene film with PMMA is first fished into deionized water with silicon wafers and left to stand for a few minutes. The graphene film with PMMA is then transferred to the target substrate and treated with natural dry. In this work, we prepared monolayer, few-layer, and multilayer graphene for temperature sensing, respectively. Different layers of graphene are used for the preparation of devices.

In experiments, graphene membranes made from CVD graphene on copper foil are suspended over cavities etched into a Si/SiO_2_ film on a silicon substrate. This CVD graphene on copper foil was gotten by wet etching. And the wet etching allows the transfer of graphene from the copper foil to the target substrate. This graphene on copper foil was purchased from Sixcarbon Tech Shenzhen. The thickness of the Si layer is 500 μm, the thickness of the SiO_2_ layer is 300 nm, and the Si/SiO_2_ substrate is from Shenzhen Rigorous Technology. The wafer is a heavily-doped monocrystalline silicon. The details of the fabrication shown in Figure 3a are as follows: (a) The Si/SiO_2_ substrate needs to be cleaned by acid cleaning and alkali cleaning to remove organic impurities on the surface of the substrate; Removal of SiO_2_ from the substrate backside using BOE. It is used for subsequent verification of the conductivity of the bottom electrode and the electrodes on two sides of the graphene. (b) A cavity is etched on the prepared sheet using the photolithography technique and RIE etching method. And the depth of the trench is 50 μm. (c) Au electrodes are sputtered on both sides of the cavity using Magnetron Sputtering equipment. (d) CVD graphene, with PMMA, is transferred onto the substrate by wetting transfer technology [8,37]. The contact of the suspended graphene with air can ensure the stability of pressure outside the cavity. Besides, the excellent heat dissipation of graphene can reduce the effect of graphene thermal resistance on the sensor. In addition, details of the wet transfer have been described earlier in this section. A PMMA film was spin-coated on the graphene on copper foil in wet transfer technology to protect the integrity of the graphene film. The conventional wet transfer will remove PMMA after transferring the graphene with PMMA to the target substrate. This increases the probability of graphene breakage [37]. However, the devices in this article do not require PMMA removal, and PMMA plays a positive effect on the device’s performance in the subsequent experiments. Therefore, the sensor prepared in this work has a high yield. Also, the highest temperature was 120 °C because the PMMA will be instability taste in higher temperatures. 

Preliminary verification of the conductivity of the sample is required because the graphene may be damaged during the transfer process. The finished samples need to select resistance values in the range of 1 kΩ–10 kΩ for subsequent testing. The device yield by the fabrication process is about 90%. Also, the suspension of graphene on the device needs to be verified. The device designed in this work can also be used as a transistor. When the graphene is not suspended but in contact with the silicon substrate at the bottom of the cavity, then the source/drain and the gate will be conductive. Then verifying the conductivity between the source and drain can be a preliminary verification of whether graphene is suspended or not. Then, the qualified samples need to be wire-bonded. In addition, the sensing structure in our work is monolayer, few-layer, and multilayer graphene. The SEM image and Raman test data of the device are shown in Figure 3b(i). Figure 3a shows the integrity of the sensitive film of the device. Intact sensitive films are very important for subsequent measurements, so qualified samples need to be selected in SEM. Figure 3b(ii) illustrates the overall thickness of the sensitive film. The expected sensitive film thickness is 500–510 nm, and this difference can be accepted by taking into account the limitations of the experimental equipment. Figure 3b(iii) shows the graphene film with PMMA in the suspended rotational state. Figure 3b(iv) shows the Raman characterization of the graphene with PMMA removed.

In our work, the graphene temperature sensor provides a direct electrical readout of temperature to strain transduction. The hot plate simulated ambient temperature. And Keithley 2611B is for monitoring current change under different temperatures. The specific measurements are as follows: (1) The temperature sensor is fixed on the hot plate with high-temperature tape; (2) The temperature was set as 25 °C. Then, save the I-V curve data after the device value becomes stable. (3) Increase the temperature of the hot plate to 120 ℃ and save the I-V data after the temperature reaches stability. And the temperature is changed in steps at 5 °C. (4) Export the test data and analyze the sensing property between graphene resistance and temperature. Finally, the testing data changed from I-V to R-V data for the benefit of the analysis.

## 3. Results

The suspended graphene sensors data in Figure 4b,d,f were first compared to blank control. The blank control, fabricated in parallel but without cavities, is adopted to verify the presence of the cavity leading to the mechanical bending and straining of the membrane that causes the pressure dependence of the resistance. The blank control’s data is shown in Figure 4c,e,g. The device output resistance of the blank control group showed little dependence on the temperature change. This is in line with expectations. In the test section, each sample was tested I-V at 1 V,2 V,3 V,4 V, and 5 V supply voltage and finally the test data was converted to R-V. There is only one curve at 1 V in Figure 4f due to the limitation of the device resistance causing the cutoff voltage below 2 V. Figure 4b,d shows the dependence of the resistance on the voltage. This is owing to the different thermal resistance effects of graphene at different voltages [38]. However, the thermal resistance effect of graphene will lead to unstable temperature-sensitive behavior in graphene temperature. [24]. That may contribute to the outlier curves in Figure 4b–e at 2 V, in Figure 4e at 1 V, et al. Nevertheless, further studies regarding stability are required in the experiment. This is also what we need to focus on in our future research work.

The resistance does not show a regularity with temperature on non-cavity samples in Figure 4c,e,g. Then, the data on cavities samples show different temperature-sensitive properties of in different graphene in Figure 4b,d,f. The monolayer graphene is almost no temperature sensitivity by contact interface in Figure 4b. And the few-layer and multilayer graphene have a temperature sensitivity show in Figure 4d,f. In particular, the temperature sensitivity of multilayer graphene devices prevails over that of few-layer graphene. Figure 4d,f shows that the test data between resistance and temperature are in line with the expected results. It means that the cavity structure is the core of temperature sensing. When the temperature changes, a pressure difference is present between the inside and the outside of the cavity. Then, the graphene membranes, sealing in the caves, are deflected and thus strained. Then the mechanical strain in the graphene induced a piezoresistive effect, which changes the electronic band structure. This leads to a change in device resistivity. However, resistance is affected by the thermal resistance effect, the substrate impurity scattering effect, and the piezoresistive effect. And the cavity structure attenuates the substrate impurity scattering effect and the thermal resistance effect by suspending the graphene [39]. So, the cavity structure improves the sensitivity by attenuating the error factors that affect the output resistance. As two-probe measurements were performed, the contact resistance also have a dependence on the temperature [27]. Using the Wheatstone bridge method to measure resistance will reduce the effect of contact resistance on device resistance. As the influence of the change of contact resistance on the device resistance is acceptable within the error range. There is reason why we believe that the data in Figure 3 are reliable. 

## 4. Discussion

From the data in Figure 4d,f, the few-layer and multilayer graphene temperature sensing structures exhibit good temperature-sensitive properties. And the sensitivity of multilayer graphene sensors is higher than that of few-layer graphene sensors. At different test voltages, the output resistance of both few-layer and multilayer graphene temperature-sensitive structures increases linearly with temperature. Besides, there is no obvious temperature-sensitive of monolayer graphene temperature. The resistance of monolayer graphene is limited mainly by substrate impurity scattering [29,39,40]. Such a piezoresistive effect of the graphene film has little effect on its resistance. And the influence of substrate impurity scattering decreases with the increase in the number of graphene layers. Therefore, the resistance of few-layer and multilayer graphene is limited mainly by the piezoresistive effect. 

Finally, the sensitivity and linearity of the sensor were analyzed and calculated. In resistance-based temperature sensors, the initial resistance (typically based on 0 °C) and the temperature coefficient of resistance (TCR) are essential parameters for temperature measurement [23]. A higher TCR value means higher sensitivity. In this work, TCR will be used as the sensitivity of the temperature sensor. The sensitivity of the graphene temperature sensors (TCR) expressed by Equation (3)
(3)TCR=∆RR∆T

A linear fit of the tested resistance data with temperature performed by Origin is in Figure 5. Figure 5a shows the fit of the test data of few-layer cavity graphene at 1 V, and Figure 5b shows the fit of the test data of multilayer graphene at the same voltage. The fitting results show that all the test data in (a, b) are within the confidence interval of 95%, which indicates that the fitted results have a high confidence level. Among the results, the linearity of the fitted curve for the few-layer graphene test data is 0.97. And the linearity of the multilayer graphene is 0.98 in the temperature range of 60 °C–120 °C and 0.61 in the temperature range of 35 °C–60 °C. In brief, the sensitivity of this temperature sensor to few-layer and multilayer graphene is 1.07%/°C and 3.50%/°C. Also, the temperature range is 25–120 °C. The performance parameters of graphene temperature sensors in different works [41,42,43] are listed in Table 1. This sensor’s sensitivity is an order of magnitude higher than the graphene temperature sensor, which is comparable with the works in this field. This sensor’s temperature range is also wide. Besides, this sensor uses CVD graphene, which fabricating cost is much lower and fit for large-scale production.

## 5. Conclusions

In this work, the sensor provides a direct electrical readout of temperature to resistance transduction by the nano piezoresistive effect in graphene. And the cavity structure can weaken the substrate impurity scattering and thermal resistance effect, which achieves better sensitivity and wide-range temperature sensing. The sensor’s sensitivity is 3.50%/°C, the linearity is 0.98, and the sensing temperature range is 25 °C–120 °C. Compared with conventional thermal resistance graphene temperature sensors, the sensors have orders of magnitude higher sensitivity. Further, the experiments suggest that the graphene layers affect the sensitivity, linearity, sensing range, and other sensor parameters. In the experiments, the monolayer, few-layer, and multilayer graphene sensors have different sensor parameters. This work can provide theoretical and technical support for NEMS temperature sensors.

## Figures and Tables

**Figure 1 nanomaterials-13-01103-f001:**
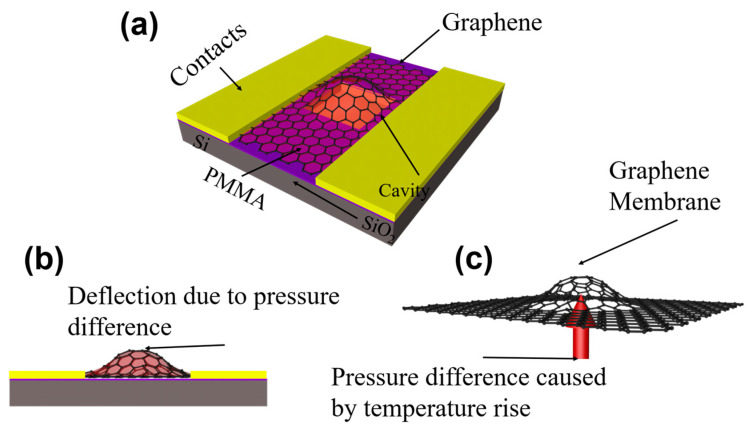
The working principle of the graphene temperature sensor. (**a**) Structure diagram of the graphene temperature sensor. (**b**) Representation of membrane functionality in a graphene temperature sensor. (**c**) Representation of sealed gas function in the cavity in a graphene temperature sensor.

**Figure 2 nanomaterials-13-01103-f002:**
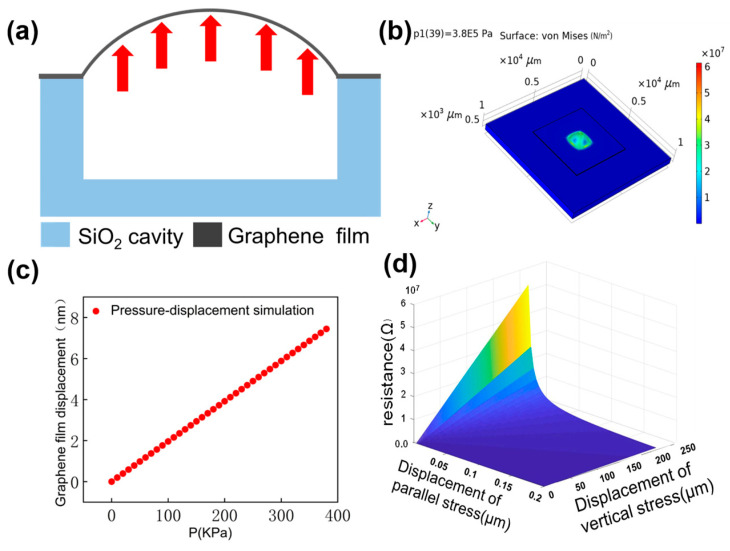
(**a**) Principle of signal conversion of sensor sensitive unit The arrows show the expansion of the film when heated. (**b**) Displacement of monolayer graphene with PMMA at 385 kPa. (**c**) Graphene film Pressure-displacement simulation results in COMSOL. (**d**) Simulation of the piezoresistive effect for predicting graphene resistance changes in MATLAB software.

**Figure 3 nanomaterials-13-01103-f003:**
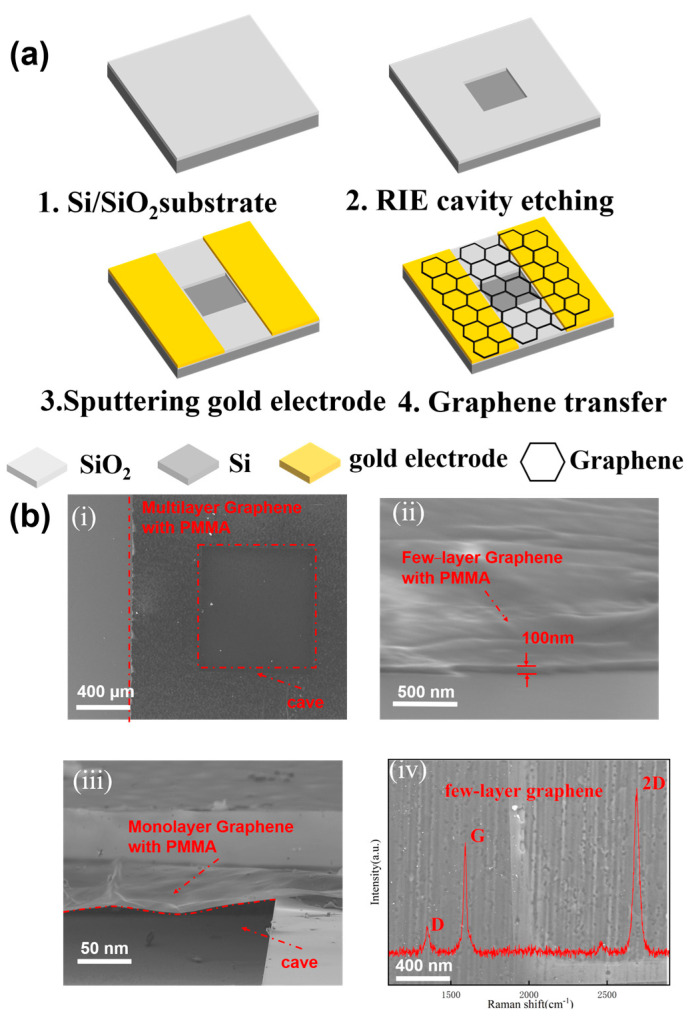
(**a**) The preparation process of the sensor. (**b**) The optical and electron microscope photos of the sensor. (i) Top view of sensor in the electron microscope. (ii) Characterization of graphene thickness. (iii) The floating state of graphene, (iv) The Raman spectrum of few−layer graphene.

**Figure 4 nanomaterials-13-01103-f004:**
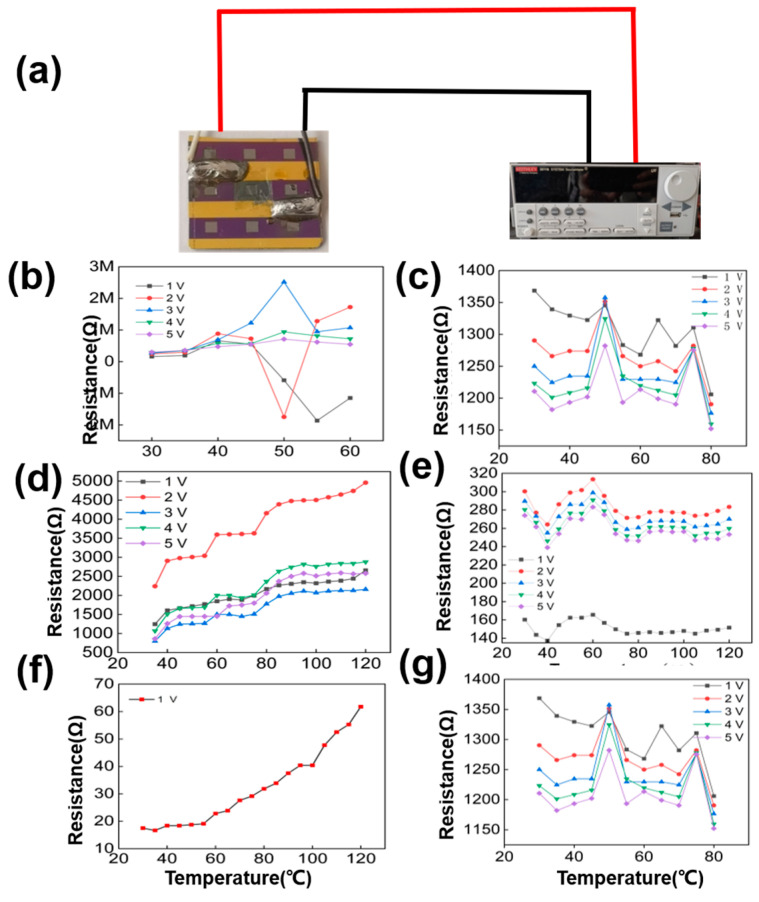
(**a**) Test data of resistance of different devices with temperature. (**b**,**d**,**f**) is the test data of monolayer/few−layer/multilayer graphene on cavity substance and (**c**,**e**,**g**) are the test data of the monolayer/few−layer/multilayer graphene on non−cavity substance. The breakdown voltage of the multilayer graphene, in (**f**), is less than 2 V due to its resistance limitation. Therefore, only 1 V bias voltage is available for this sample.

**Figure 5 nanomaterials-13-01103-f005:**
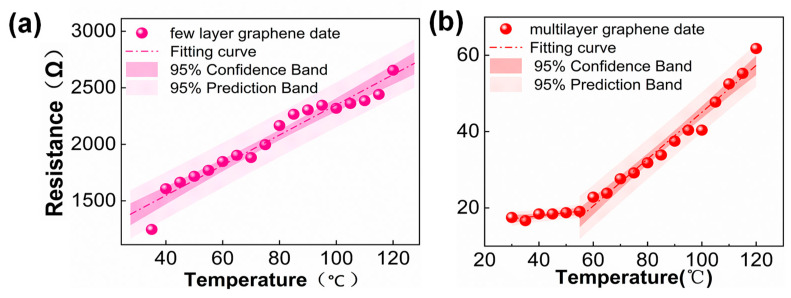
Sensitivity fitting curves for few-layer and multilayer graphene cavity structures. The sensitivity equation S = (ΔR/R)/ΔT shows that the sensitivity of the few-layer graphene cavity is 1.07%/°C and the sensitivity of the multilayer graphene cavity is 3.50%/°C. (**a**) is the sensitivity of few-layer graphene. (**b**) is the sensitivity of multilayer graphene.

**Table 1 nanomaterials-13-01103-t001:** Graphene temperature sensors in different works.

Material	Operating Rang	TCR *
BN/Gra/BN [23]	30–150 °C	0.25%/°C
Monologue Gra [21]	9–29 °C	−0.16%/°C
Muti-walled CNTs [31]	−33–126 °C	−0.13%/°C
Graphene, PP texti [32]	30–70 °C	−0.17%/°C
CNTs prepared by CVD [33]	30–150 °C	0.04%/°C
**Multilayer Gra (This work)**	**25–120 °C**	**3.50%/°C**

* TCR is the sensitivity of the graphene temperature sensor.

## Data Availability

The data that support the findings of this study are available from the corresponding author upon reasonable request.

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
