# Peer review of "Nanoelectromechanical Temperature Sensor Based on Piezoresistive Properties of Suspended Graphene Film"

_nanomaterials, 2023, doi:10.3390/nano13061103_

Round 1

Reviewer 1 Report

This paper deal with monolayer, few-layer, and multilayer graphene on the cavity and non-cavity SiO2/Si substrate. The authors show that such structures, used as sensors, provide temperature-to-resistance transduction by the nano piezoresistive effect. Furthermore, it is shown that the cavity structure with few-layer and multilayer graphene results in higher sensitivity and wide-range temperature sensing.

Although not novel, the study could be interesting but its scientific strength must be enhanced.

The paper can be reconsidered for publication after a substantial revision. Improvements and clarifications are needed as detailed below:

-          Improve English in sentences such as “In addition, monolayer graphene is almost no temperature sensitivity by contact interface.”, “The temperature change, a pressure difference, is present between the inside and the outside of the cavity”, etc.

-           “Its maximum detectable temperature is up to 2100 °C.” Where does this number come from? Add a reference.

-          “However, the substrate impurities scattering of graphene will lead to unstable temperature-sensitive behavior and poor linearity in graphene temperature sensors.” The authors could add that sometimes a change from negative to positive temperature coefficient of resistance is observed for graphene on SiO2 (see for instance https://doi.org/10.1088/2632-959X/ab7055 that can be added as reference).

-          “In summary, the thermal resistance graphene temperature sensor suffers from the drawbacks of enhancing the sensitivity and widening the temperature sensor range.” Rewrite this sentence that is ambiguous. Enhancing the sensitivity and widening the temperature sensor range is not a drawback but an advantage.

-          Separate the figure captions from the text. For instance, in figure 3 it is difficult to understand where the figure caption terminates.

-          Device fabrication: Give the thickness of the SiO2 layer and the depth of the trench.  How is graphene obtained? The authors consider few-layer/multilayer graphene but no details are given.

-          “The contact of the suspended graphene with air can ensure the stability of pressure outside the cavity. Besides, the excellent heat dissipation of graphene can reduce the effect of graphene thermal resistance on the sensor. The highest temperature was 120 °C because the PMMA will be instability taste in higher temperatures. In this work, PMMA is used for protecting graphene membranes.” I am not sure I understand this part. Please make it clearer. If I understand correctly a PMMA layer is left on top of graphene. This layer can have non-negligible effects on the graphene bending and electrical properties, which have not been considered in the COMSOL simulation.

-          There is no proof that the authors are studying “suspended graphene” as claimed.

-          “The device yield by the fabrication process is about 90%.” What about device reproducibility?

-          “The suspended graphene sensors date in Figures 4 (b), …” -> Is it data?

-          The data in figure 4 needs clarification. I guess that the different curves are R-T curves at different biases between the Au contacts. Why is there a dependence on the voltage? And why is it non-monotonic? For instance, in c), why is the curve at 2V higher than the other ones?

-          “The resistance does not show a regularity with temperature on non-cavity samples. Then the data on cavities samples show different temperature-sensitive properties of indifferent graphene.” What does “regularity” mean here? I do not see regularity in plot b) which is referred to cavity substance. I guess it is “different graphene”.

-          “This leads to a change in device resistivity. How-170 ever, resistance is affected by the thermal resistance effect, the substrate impurity scattering effect, and the piezoresistive effect. And the cavity structure attenuates the substrate impurity scattering effect and the thermal resistance effect by suspending the graphene.” What about the effect of contact resistance? As two-probe measurements are performed, the contact resistance can have a dependence on temperature, which the authors should consider (refer for instance to the above mentioned paper https://doi.org/10.1088/2632-959X/ab7055). Also, it is not clear to me why the cavity structure attenuates the thermal resistance effect.

-          “At different test voltages, the output resistance of both few-layer and multilayer graphene temperature-sensitive structures increases linearly with temperature.” Is this behavior expected? Why? Is it confirmed by the literature?  

Reviewer 2 Report

Authors studied the temperature sensing of chemical vapor deposited few, multi layered graphene. The cavity in the substrate SiO2 helps to enhance the piezoresistive effect which resulted in the improved sensitivity of the temperature sensor of the multi layered graphene. It would be nice if the below points are addressed.

  1. Below published literature, I have found in a quick google search on the same topic, where the effect of cavity studied in detail on NEMS devices and pressure sensor. Authors might clearly state what are the new findings obtained in this work.
    1. Mohammad Haniff, M., Muhammad Hafiz, S., Wahid, K. et al. Piezoresistive effects in controllable defective HFTCVD graphene-based flexible pressure sensor. Sci Rep 5, 14751 (2015).  https://doi.org/10.1038/srep14751
    2. S. Wagner, C. Weisenstein, A.D. Smith, M. Östling, S. Kataria, M.C. Lemme, Graphene transfer methods for the fabrication of membrane-based NEMS devices, Microelectronic Engineering, 159, 108-113 (2016). https://doi.org/10.1016/j.mee.2016.02.065.
  2. Page-5 results section, Figure 4 (b) not discussed in the text.
  3. Page-5 results section, there is a typo 'data' is written as 'date' (lines 155, 158)
  4. The manuscript needs to be reviewed carefully for the spelling/typo.
  5. Page-6 Table-1 mentioned a abbreviation TRC, which is not defined/described in the text.
  6. The preparation conditions for few, multi layer graphene could be included in the experiments section.

Reviewer 3 Report

The Piezoresitive properties of suspended graphene films presented in this work is very interesting and does add value to the scientific gap in the research of graphene-based temperature sensors. Therefore, the article could be considered for publication after minor adjustments based on the following points: 

1. Have authors considered how the type of the cavity impact the sensitivity?

2.Correct way of in-text refencing must be taken into consideration, i.e. the use of "et.al." or ".... and co-workers".  Moreover, the references section must be corrected.

3. Lines 77-84 seem to be concluding the data before the presentation of the results, thus must be moved to the appropriate section.

4. The Experimental section needs detailed clarification.

5. Line 112, should be eqn (2) instead of eqn (3).

6. Authors claimed the choice of the experimental conditions was such that they were within the tearing limits of graphene; thus, what are the typical tearing limits of graphene?

7. Subsection 2.1 Experimental must be corrected to 2.2.

8. Which type of CVD technique was used for the synthesis of graphene?

9. Full Raman spectrum is recommended to ascertain complete lack of defects on the graphene films.

10. Line 132, reference needed for the wet transfer technique.

11. Have authors taken into consideration the influence of the thermal and strain properties of PMMA ? Why wasn't hBN used for encapsulation?

12. Why was the cavity/multilayered graphene only tested with one voltage?

13. Lines 155 and 158, the word should be "data"  and not "date".

14. Contributions of each and every listed authors must be detailed. Also the details of the corresponding authors are missing.

Round 2

Reviewer 1 Report

I appreciate the attention given to my comments. The authors satisfactorily addressed most of my comments. However, there are a couple of points that require more elaboration.

First, I suggest avoiding sentences such as “In this thesis,…”. This is supposed to be a scientific article, not a student thesis, although it stems from a student thesis.

To demonstrate that the graphene is suspended the authors say “When the graphene is not suspended but in contact with the silicon substrate at the bottom of the cavity, then the source and drain regions will be conductive. Then verifying the conductivity between source and drain can be a preliminary verification of whether graphene is suspended or not.” Source and drain are connected by graphene, so there will be an electrical current for both suspended and not suspended graphene. This requires further clarification.

Also, the authors attribute the dependence of resistance on the voltage to the thermal resistance effect of graphene. This is not very convincing. Although the authors honestly admit that such an issue requires more studies, they copuld elaborate a bit more.

Reviewer 2 Report

  1. Authors didn't address the significant difference of this work compared to below published literature, I have found in a quick google search on the same topic, where the effect of cavity studied in detail on NEMS devices and pressure sensor.
    1. Mohammad Haniff, M., Muhammad Hafiz, S., Wahid, K. et al. Piezoresistive effects in controllable defective HFTCVD graphene-based flexible pressure sensor. Sci Rep 5, 14751 (2015).  https://doi.org/10.1038/srep14751
    2. S. Wagner, C. Weisenstein, A.D. Smith, M. Östling, S. Kataria, M.C. Lemme, Graphene transfer methods for the fabrication of membrane-based NEMS devices, Microelectronic Engineering, 159, 108-113 (2016). https://doi.org/10.1016/j.mee.2016.02.065.
  2. Authors have mentioned "in this thesis.." at two places line-119 and 149.
  3. The manuscript need to be thoroughly checked for any mis-spelling/typo. For instance line 175-176 "This CVD gra-175 phene on copper foil was getten by wet etching."
  4. Page-6 Table-1 mentioned a abbreviation TRC, TRC's full form is not mentioned in the text.

Round 3

Reviewer 1 Report

I appreciate the attention given to all my comments and suggestions. The authors made changes and improvements in their manuscript and also corrected some technical errors. 

The revised version of the manuscript is technically sounder. The paper can be accepted for publication in its current form.

Reviewer 2 Report

  1. Thanks to Authors for addressing the comments.

    1. TRC (Temperature resistance coefficient) is changed to TCR (Temperature coefficient of resistance) in the text, still the Page-6 Table-1 mentioned a abbreviation TRC. It would be more clear to keep either TRC or TCR with the clear description.
    2. I would recommend to double check the manuscript thoroughly for any mis-spelling/typo.
